# Practical Approach to Hypersensitivity to Nonsteroidal Anti-Inflammatory Drugs (NSAIDs) in Children

**DOI:** 10.3390/ph16091237

**Published:** 2023-09-01

**Authors:** Daniela Podlecka, Anna Socha-Banasiak, Joanna Jerzynska, Joanna Nodzykowska, Agnieszka Brzozowska

**Affiliations:** 1Department of Pediatrics and Allergy, Copernicus Memorial Hospital in Lodz, Medical University of Lodz, 90-419 Lodz, Poland; daniela.podlecka@umed.lodz.pl (D.P.); joanna.jerzynska@umed.lodz.pl (J.J.); jnodzyk@gmail.com (J.N.); 2Department of Gastroenterology, Allergology and Pediatrics, Polish Mother’s Memorial Hospital Research Institute, 93-338 Lodz, Poland; sochabanasiak@gmail.com; 3Military Medical Academyy Memorial Teaching Hospital of the Medical, University of Lodz-Central Veteran’s Hospital, 90-549 Lodz, Poland

**Keywords:** NSAIDs, hypersensitivity, angioedema, urticaria, children

## Abstract

Background: We aimed to assess the real-life prevalence, patient profile, and clinical presentation of drug hypersensitivity to NSAIDs in children after an incidence of an adverse event during treatment, verified by a drug challenge test. Methods: We included 56 children, aged 4–18 years, referred to our allergy clinic due to the incidence of adverse reaction during treatment. Skin prick tests and a drug provocation test were performed in all patients. Diagnostics for persistent urticaria were performed. Results: In 56 patients suspected of drug allergy, we proved NSAID hypersensitivity in 17 patients (30.1%). In 84.9% (*n* = 47) of patients, the clinical manifestations of hypersensitivity revealed angioedema and urticaria. The most common culprit drug among NSAIDs in children was ibuprofen. Thirty-one (55.4%) reactions were immediate, and 25 (44.6%) were delayed or late. Previous history of allergy was a risk factor for NSAID hypersensitivity (*p* = 0.001). Vitamin D deficiency in the blood serum was a risk factor for NASID hypersensitivity (OR = 5.76 (95% Cl: 1.42–23.41)). Conclusions: Hypersensitivity to NSAIDs is a difficult diagnostic problem in pediatric allergy. The most common manifestation of hypersensitivity to ibuprofen in children is acute urticaria and angioedema. Two important problems in the differential diagnosis are cofactors such as vitamin D levels and viral infections, which require further research.

## 1. Introduction

According to the World Health Organization, an adverse drug reaction in adults and children is defined as “any harmful, unintended and undesired effect of a drug that occurs at doses used for treatment, prevention or diagnoses” [1]. The majority of these reactions are categorized as type-A reactions, which are described as predictable, common, usually dose-dependent, and caused by previously known pharmacological characteristics of the drug and its side effects [1,2,3]. Drug allergies are categorized as type-B reactions, which are supposed to be independent of dose and affect a small population, which suggests that individual patient factors are key [2]. Nonsteroidal anti-inflammatory drugs (NSAIDs) are one of the most often administered drugs within the pediatric population, mainly as anti-inflammatory, analgesic, and antipyretic medications [1,2]. Nonsteroidal anti-inflammatory drugs are the most frequently given drugs in infections as painkillers, anti-inflammatories, or antipyretics. In accordance with the FDA recommendations and the characteristics of medicinal products, paracetamol, ibuprofen, or metamizole are allowed for use in the youngest children as analgesics, anti-inflammatories, or antipyretics. Acetylsalicylic acid is only approved for use in children over 12 years of age due to the probable appearance of Reye’s syndrome. Due to the above, in the presented study, cases of hypersensitivity after the above-mentioned four drugs were analyzed. Of course, in special situations, such as severe pain, especially of rheumatoid origin, or with no alternative, naproxenum can be used in children over 5 years of age. We usually use naproxen as an alternative drug in patients with confirmed allergies to paracetamol and ibuprofen. The prevalence of NSAID hypersensitivity is reported to be approximately 2–6% in the general population; however, exact data concerning children is lacking [3]. Nevertheless, many studies point to NSAIDs as the second-leading drug group responsible for evoking hypersensitivity reactions following beta-lactam antibiotics [4,5,6]. In particular, paracetamol and ibuprofen are the most often reported NSAID agents in children, largely due to the fact that these drugs are approved for use in this age group [7,8,9].

The mechanism of hypersensitivity to NSAIDs is non-immunological and attributed to inhibition of cyclooxygenase (COX), the enzyme that metabolizes arachidonic acid, resulting in the synthesis of prostaglandins, thromboxanes, and prostacyclins [10]. COX—which is inhibited by NSAIDs—exists in two isoforms, COX-1 and COX-2. COX-1 is constitutively expressed in most cells, and its activation leads to the synthesis of prostanoids, which play an important role in platelet aggregation. Meanwhile, COX-2 is expressed both constitutively and is induced by inflammatory stimuli, and is responsible for the synthesis of pro-inflammatory prostanoids [10]. According to ENDA/EAACI and its position paper, the majority of NSAID hypersensitivity cases in children up to 10 years of age are thought to be due to non-immunological responses, which are often triggered by infection or physical activity [8]. In younger children, viral infections are often linked with maculopapular rashes and urticaria, which makes it all the more difficult to distinguish from allergy. In older children, the mechanisms are the same as in adults, and can involve immunological and non-immunological pathways.

According to the literature, the most common symptoms of hypersensitivity to NSAIDs are maculopapular eruptions, urticaria, and angioedema [11]. The differential diagnosis of hypersensitivity to NSAIDs and urticaria is difficult and very broad, as it involves carrying out numerous tests with the aim of showing different potential causes of urticaria other than the drug used. Standard diagnostic tests for urticaria and angioedema should include the use of medications and dietary supplements (new or recently changed doses), allergies, recent infections, travel history, family history of urticaria and edema, history of infectious diseases such as viral hepatitis, Ebstein-Barr, human immunodeficiency virus, and vitamin and micronutrient deficiencies [12].

According to the review of the EAACI/ENDA and GA2LEN/HANNA, urticaria and angioedema may be induced by various NSAIDs in healthy individuals. Additionally, challenge studies with other COX-1 inhibitors have shown that as many as 30% of patients with a history of hypersensitivity to NSAIDs are sensitive to just a single NSAID [13]. According to current knowledge, hypersensitivity to NSAIDs can be described in relation to a single drug (selective reactor (SR)) when the clinical symptoms are caused by a single NSAID drug with concomitant good tolerance to other drugs. Generally, the term includes allergic hypersensitivity reactions to NSAIDs. Two phenotypes of selective hypersensitivity responses to NSAIDs have been defined. The first is selective urticaria, angioedema, and/or NSAID-induced anaphylaxis (SNIUAA)—immediate reactions, possibly mediated by specific IgE antibodies. The second group is selective delayed selective hypersensitivity-type reactions induced by NSAIDs (SNIDR)—reactions occurring within 24–48 h after ingestion of the drug, probably mediated by a specific T-cell response.

The aim of this study was to assess the real-life prevalence, patient profile, and clinical presentation of drug hypersensitivity to NSAIDs in children referred to our allergy clinic after experiencing an incidence of adverse effects during treatment, verified by a drug challenge test.

## 2. Results

The current analysis is restricted to 56 children (boys and girls), aged 4 to 18 years, who underwent a full diagnosis of suspected drug allergy. Patients with chronic diseases requiring prolonged drug intake were excluded from the study. Patients with severe drug allergy reactions were excluded from the study, as a severe drug allergy reaction is a contraindication for drug provocation testing.

Any mental disorders limiting patient contact and compliance were also exclusion criteria. Baseline characteristics are given in Table 1.

The most commonly reported culprit drug was ibuprofen (N = 33 cases; 11 positive via DPT), followed by paracetamol (N = 13 cases; 1 positive via DPT), ASA (N = 8 cases; 4 positive via DPT), and metamizole (N = 2 cases; 1 positive via DPT).

From all 56 patients suspected of drug allergy, after all diagnostic procedures were carried out and analyzed, the results indicated hypersensitivity to NSAIDs in 17 patients (30.1%). In 84.9% of patients (*n* = 47) the clinical manifestation of symptoms, considered a possible hypersensitivity reaction, was observed as angioedema and urticaria. Clinical manifestations of severe anaphylactic reactions were recorded in two cases. Regarding the time of clinical manifestation of reactions, 55.4% (*n* = 31) were immediate, and 44.6% (*n* = 25) were delayed or late reactions. Positive provocation test results in the study’s participants were not related to their age (*p* = 0.3337).

In our group of patients, symptoms suspected of being due to hypersensitivity occurred after oral administration in 48 cases, after rectal administration (suppository) in 4 cases, and after intravenous administration during infection in 2 cases.

Previous history of allergy in the studied patients, as well as previous history of allergic conditions within their families, were risk factors for drug allergy (*p*-value 0.001) (Table 2 and Table 3).

The predictive validity indicators of the provocation diagnostic test were as follows: sensitivity, 65.4%; specificity, 100%; positive predictive value, 100%; negative predictive value, 76.9%; and disease prevalence, 46.4%. We also found a link between female gender and a higher incidence of NSAID hypersensitivity (Table 4).

Vitamin D deficiency in blood serum was a risk factor for NSAID hypersensitivity in the studied group (OR = 5.76 (95%Cl: 1.42–23.41) (Table 5).

During the diagnosis of latent infections as a potential cause of urticaria and angioedema, it seemed that a significant proportion of the studied patients had recently been infected with EBV; however, in the study group, there was no statistically significant relationship with hypersensitivity to NSAIDs, as confirmed by a positive challenge test (*p* = 0.4317).

All other analyses for latent infections or parasites were not statistically significant.

## 3. Discussion

The factors that determine the type and presentation of NSAID hypersensitivity reactions in children are still unknown. Hypersensitivity to NSAIDs is not limited to only oral administration of the preparations. This is the most common route of administration, but there are cases of hypersensitivity with rectal and/or intravenous administration. Although, in many cases, the reactions to NSAIDs are mild and not life-threatening, they limit future administration of these drugs to affected children [8]. Drug allergy not only affects the patient’s quality of life but may also lead to malpractice and the use of suboptimal alternative medications. According to data derived from the literature, the incidence of hypersensitivity to NSAIDs is lower in children than in adults; however, hypersensitivity reactions to NSAIDs in children appear to be more common than previously thought [8]. In part, this may be due to the overall increasing prevalence of allergic diseases [8,9,13]. Of the 56 children referred to our clinic who were “labeled” as having hypersensitivity to NSAIDs, we confirmed only 17 cases of allergy (30.4%) after analyzing DPT results; other reactions were non-specific and most probably linked to the patient’s main underlying disease. The most often reported culprit drug was ibuprofen, which is consistent with the results of other studies [6,8,14]. This seems obvious, as ibuprofen is confirmed to be the drug of first choice in the treatment of inflammatory pain in children, and, as a result, it is the most commonly prescribed anti-inflammatory drug in children. An important topic highlighted by Cravidi C. et al. is the fact that, in the case of a hypersensitivity reaction to a NSAID, clinicians should first assess if the patient presented the reaction only to one molecule or to more [15].

Drug hypersensitivity can be induced by various mechanisms. Elsagallai et al. [16] showed that a drug or its metabolites can act as a hapten that is able to covalently bind to a protein, and the resulting combination can induce allergic reactions mediated by IgE or T lymphocytes. Moreover, drugs can also directly stimulate receptors such as HLA or T-cell receptors (TCR) [17]. Clinical observations suggest that viral infections also promote or aggravate drug-related skin rashes [18].

In the study, in approximately 85% of the patients, the clinical manifestations of NSAID hypersensitivity manifested as angioedema and urticaria. The primary mechanism of angioedema and urticaria is the release of histamine and other mediators from mast cells and basophils. When the release of mediators occurs in the dermis, it clinically manifests as urticaria, whereas when the release occurs in the deeper layers of the dermis or subcutaneous tissues, angioedema occurs. Most often, basophils and mast cells are stimulated to release inflammatory mediators by IgE mediation, but other mechanisms are also known, such as non-IgE and non-immune activation of mast cells. According to Milosecic K. et al., the most common manifestation of drug hypersensitivity in children was acute urticaria (78.2%), followed by exanthema (10.5%) and angioedema (5.3%) [19].

Paracetamol and ibuprofen are the most frequent elicitors of SNIUAA in children [20]. In our study, most of the patients manifested urticaria and angioedema, mostly as immediate reactions after NSAID administration; however, SPT in all cases was negative in our patients. It is worth mentioning that among all patients with positive drug challenge tests, most of the patients had symptoms labeled as hypersensitivity during an on-going infection, and just two of these patients had used NSAIDs as painkillers without simultaneous symptoms of infection.

In a recent study by Kuhl et al. that analyzed cofactors of drug hypersensitivity, a positive association of antibiotic hypersensitivity with obesity was found, and age was negatively associated with arterial hypertension, female sex, elevated immunoglobulin E, and allergic rhinitis [21]. Hypersensitivity to NSAIDs was associated with atopic dermatitis, elevated IgE, and arterial hypertension. Our study showed a previous history of allergy in the studied patients, and also a previous history of allergic family conditions were risk factors for drug allergy in the studied patients (*p*-value 0.001). What is more, in the pediatric population, especially in children < 7 years of age, most drugs are administered in the form of syrups or suspensions. In drugs prepared in this way, apart from the active substance, other ingredients are also present, such as, for example, a flavoring agent, an emulsifier, a stabilizer, a preservative, and so on. In practice, any of these substances could cause a hypersensitivity reaction. Therefore, during the oral challenge test of the diagnostics, we used a powdered and measured part of the drug with the aim of administering the purest form of the drug.

Vitamin D plays an important role in reducing inflammation through various mechanisms, such as inhibiting the function of B lymphocytes and the secretion of immunoglobulin E (IgE), reducing the secretion of cytokines from Th1 lymphocytes, inhibiting the production of Toll-like receptors, and increasing the production of interleukin-10 in mast cells [22]. It is known that maintaining the well-being of human skin, which is a significant barrier, plays a role in the prevention of skin allergies [23,24]. The human immune system is characterized by the ability to recognize previously known pathogens and adapt its actions to them. Almost all cells of the specific immune system express the receptor for vitamin D, which makes them sensitive to the presence or absence of a stimulus such as the concentration of vitamin D. It is known from the literature that vitamin D inhibits the proliferation of T lymphocytes by reducing the secretion of Th1 cytokines. Additionally, foreign antigens are recognized by B lymphocytes, which activate them and then transform them into antigen-producing plasma cells and memory cells. It has also been shown that vitamin D affects the functions of B lymphocytes and modulates the humoral immune response, including the secretion of immunoglobulin E (IgE) [25]. Vitamin D enhances defense mechanisms within the skin barrier and generally decreases inflammation in the innate and adaptive immune systems [26]. Several studies have shown associations between vitamin D levels and sensitization to food and aeroallergens [23,27]. Several studies have shown that serum 25(OH)D levels are significantly lower in patients with chronic spontaneous urticaria than in controls [28,29,30]. Woo et al. found that serum 25(OH)D levels were significantly reduced in patients with chronic urticaria but not in acute urticaria [31]. In our study group, we observed that low vitamin D status was linked with cutaneous symptoms perceived as an expression of hypersensitivity to NSAIDS. However, based on our findings, we are unable to determine whether vitamin D is directly linked with drug hypersensitivity or with urticaria (chronic and acute) itself. Our results indicate that low serum vitamin D status may be a risk factor for hypersensitivity to NSAIDs. According to pediatric recommendations, each child should be supplemented with vitamin D. However, it is worth taking into account the type of diet in the child and the individual absorption of vitamin D in each patient, and testing the level of vitamin D, for example, at annual check-up visits. In addition, in children with a history of hypersensitivity, even if it was not confirmed by an oral challenge test, it is worth using drugs in the form of tablets to avoid additional potentially allergenic substances. In patients with confirmed hypersensitivity to one drug from the group of NSAIDs, it is advisable to perform an oral challenge test with an alternative drug in order to find a safe drug for the child.

In the presented study, hypersensitivity to NSAIDs was confirmed statistically more often in girls compared to boys (*p* = 0.046). Data on the relationship between drug allergies in children and gender are currently lacking in the literature. Eaddy et al., in their study, showed drug allergies to be more frequently reported in adult females compared with adult males [32]. Adult females are also more likely to have drug-induced anaphylaxis. Studies in children suggest the reverse picture, with prepubertal males more likely to have drug allergies and drug-induced anaphylaxis than prepubertal girls. Possible explanations for women’s predisposition to drug allergies are rather multifactorial and may include greater exposure to antibiotics or drugs, X-linked genetic factors, epigenetic changes, and divergent hormonal interactions with immune cells. Additionally, in a study by Sousa-Pint et al., the prevalence of self-reported drug allergies was higher in female patients, adults, and inpatients.

Non-immediate drug-induced cutaneous reactions such as maculopapular exanthema, delayed urticaria, erythema multiforme, and toxic epidermal necrolysis may result from selective T-cell-mediated hypersensitivity [13]. Maculopapular eruptions are among the most common delayed cutaneous adverse reactions to NSAIDs. A similar clinical picture is often observed in the pediatric population in the course of various viral infections [33]. Viruses often induce T and B cell responses, generating specific lymphocytes and antibodies [34], whereas most drugs involved in drug-induced cutaneous reactions do not induce humoral responses [33].

On the other hand, in the pediatric population, cutaneous symptoms, especially maculopapular eruptions, are the most frequently reported symptoms [1,35,36]. According to the literature, viral exanthemas can appear during treatment of a viral infection with different drugs and can also mimic drug exanthemas in 10% of cases [18]. Viruses such as Epstein–Barr virus (EBV), human herpes virus type 6 (HHV6), cytomegalovirus (CMV), and also the bacterium Mycoplasma pneumoniae can cause rash as a result of the infection itself (active or latent) or as a result of interactions with drugs that are taken simultaneously [18]. In the study group, no significant correlation between EBV IgG levels and symptoms considered to be hypersensitivity to NSAIDs was found; however, this may be due to the small study group size and warrants further investigation. According to the standards for management of urticaria, diagnostics of latent infections are supposed to be performed in patients with chronic urticaria. In our patients, acute urticaria was observed, likely triggered by NSAIDs.

Another result that should be emphasized is the fact that a previous history of allergy in our study group and a previous history of allergic conditions in patients’ families were risk factors for drug allergy. This is consistent with Santchez et al. [37], who state that the prevalence of atopy is increased in challenge-proven NSAID-intolerant patients, and the atopic conditions may represent an important risk factor for developing reactions to these drugs. Additionally, Waheed et al. state that “drug allergy is a product of the interplay of unique factors related to a patient and a drug” [34]. Furthermore, this study also found that female individuals were more likely to have drug allergies. Similarly, Sousa et al. report that the prevalence of self-reported drug allergies is higher in female patients, especially in adults [38].

A key limitation of the study is the relatively small size of the group studied; however, we selected patients without chronic comorbidities requiring other treatment in order to avoid drug interactions. It would also be interesting to investigate the influence of other common viruses on hypersensitivity. No data on severe late reactions were collected, as the studied group didn’t include such patients.

On the other hand, the methods by which the study was conducted provided a wide spectrum of differential diagnoses with which to discover other potential causes of urticaria and angioedema in the study group. The diagnosis of drug allergies is difficult, and hastily classifying NSAID side effects as an allergy results in limiting the possibility of using analgesics and anti-inflammatory drugs in such patients. Diagnostic tools for drug hypersensitivity are currently limited, and the oral challenge test remains the diagnostic gold standard. Unfortunately, this is an invasive procedure with the possibility of causing anaphylaxis due to the fact that it involves administration of a potentially allergenic drug to the patient and requires careful observation of possible symptoms. The symptoms of anaphylaxis in children can be difficult to notice; therefore, such diagnostics should be carried out strictly by an experienced team and in intensive care unit facilities.

## 4. Materials and Methods

### 4.1. Patients

The study group included 56 children from our pediatrics and allergy outpatient clinic, aged 4 to 18 years, with a history of adverse reactions during treatment with NSAIDs in the past 6 months. Adverse reactions after NSAIDs were defined as incidences of urticaria, angioedema, maculopapular rashes, anaphylaxis, and anaphylactic shock that appeared after NSAID intake. Immediate reactions were considered reactions with an onset of clinical symptoms within 1 h after the last drug intake, in accordance with the International Consensus on Drug Allergy [39]. Reactions with an onset of symptoms after an hour following the last intake but up to 24 h were considered delayed; reactions occurring later than 24 h from the last drug intake were defined as late [39].

The study was conducted between January 2019 and July 2021 and consisted of two visits.

The study was approved by the Medical Ethics Committee of the Medical University of Lodz; RNN/147/18/KE. All parents or legal guardians gave their oral and written consent for the evaluation of data from their children’s medical documentation.

During the first visit to the pediatrics and allergy outpatient clinic, demographic characteristics and medical history concerning family history of allergy, chronic diseases, and previous treatment with NSAIDs were recorded and analyzed. Medical history regarding allergies to drugs and drugs used in the past was taken in accordance with the form developed by the United States Food and Drug Administration (FDA) Adverse Event Reporting System (FAERS). During this visit, patients were advised to cease antihistamine drug intake for a period of 14 days, at which point they would be admitted to the clinic to continue to the next part of the diagnostics. All patients were examined by a medical doctor in order to exclude ongoing infections or skin changes that could potentially cause difficulties in symptom interpretation during the provocation test. Patients with chronic diseases requiring prolonged drug intake were excluded from the study.

Children with suspected severe late hypersensitivity reactions to NSAIDs, such as Stevens–Johnson syndrome (SJS), toxic epidermal necrolysis (TEN), drug-induced hypersensitivity syndrome (DIHS), and drug reaction with eosinophilia and systemic symptoms (DRESS), were excluded from the study due to the fact that a confirmatory test in the form of an oral challenge test with a putative drug could be life-threatening [40].

During the second visit—after 14 days of discontinued antihistamine intake—further diagnostics were carried out. During the diagnosis of hypersensitivity to NSAIDs, the following tests were applied:-Physical examination performed by a medical doctor, including measurement of vital signs (heart rate, blood pressure, measurement of blood oxygen saturation);-Skin prick tests (SPT).

Keeping in mind the fact that hypersensitivity to NSAIDs is, by definition, non-immune and true IgE-mediated type I allergic reactions (single NSAID-induced urticaria/angioedema or anaphylaxis, SNIUAA) are rare, it was decided that SPTs would be performed on all patients in accordance with standard guidelines. Unfortunately, commercially validated in vitro tests to determine IgE against NSAIDs are currently not available.

A SPT, performed according to accepted procedures [41], was conducted by pricking the skin percutaneously with a prick needle through an allergen solution (drug), as well as positive (histamine phosphate 10 mg/mL) and negative (saline buffer/50% glycerol) controls. After 15–20 min of application, the tests were interpreted, with a positive result defined as a wheal ≥ 3 mm in diameter. Concentrations used for the skin prick tests were prepared according to the ENDA/EAACI Allergy Interest Group [42].

### 4.2. Drug Provocation Test (DPT)

Due to the fact that the investigated population consisted of children with considerable differences in weight, protocols described by Zambonino et al. [41,43] were used to unify the threshold dose—3 steps: ¼, ¼, and ½ of the total cumulative dose (paracetamol, 15 mg/kg/dose; ibuprofen, 10 mg/kg).

For acetylsalicylic acid (ASA), a protocol developed by Nizankowska [44] was implemented.

Each DPT lasted 2 days—day 1: placebo only, and day 2: drug testing. Subsequent doses were administered every hour.

Before each dose of placebo/DPT, vital signs (heart rate, blood pressure, measurement of blood oxygen saturation, and spirometry, when appropriate) were performed. The DPT was considered positive if objective signs occurred during drug administration. In all cases in which subjective symptoms appeared, the supervising physician could decide either to repeat the last dose or to divide the next dose into 2 steps. When a patient reported subjective symptoms but completed the DPT without objective signs, the DPT was considered negative. When a reaction interpreted as positive appeared at any time during DPT, the DPT was stopped and the condition was treated immediately.

In accordance with ENDA/EAACI recommendations for DPT indications, we performed DPTs after ensuring all safety measures and precautions (intravenous catheter, prepared emergency set) were taken in case of anaphylaxis during DPT [45]. The DPT was carried out by using uniform capsules delivered in specified doses prepared by the hospital pharmacy.

As a result of most of the patients manifesting symptoms of urticaria and angioedema after NSAID intake, they also had diagnostics for chronic urticaria. All patients had blood and stool samples taken in order to check for latent viral or bacterial infections and parasites. Blood samples were performed for routine screening of enzyme immunoassay with enzyme-linked immunosorbent assays (ELISA) for IgG and IgM antibodies against Ascaris lumbricoides, Toxocara canis, Toxoplasma, Ebstein–Barr virus, cytomegalovirus (CMV), hepatitis C virus, Mycoplasma pneumoniae, and concentration of vitamin D (25(OH)D) in the serum. Stool samples were investigated for the presence of potent parasite eggs and cysts and Helicobacter pylori antigen. All tests were performed in a hospital laboratory in accordance with the manufacturer’s instructions.

Infections with Ascaris lumbricoides or Toxocara canis were considered positive when appropriate IgG antibodies were >11.0 NTU. For Toxoplasma gondii IgG antibodies, the cut-off was ≥30 IU/mL.

Recent EBV infections were recorded if EBV IgG S/CO levels were defined as reactive when ≥1.00 S/CO (nonreactive: <0.75; gray zone: 0.75–1.00) and for CMV recently underwent infection ≥ 1 IU/mL. Infection with HCV was recorded if anti-HCV antibodies were found.

Infection with Helicobacter pylori was considered if an antigen against Helicobacter was found in the stool sample.

Vitamin D deficiency was defined as 25(OH)D levels less than 20 ng/mL.

### 4.3. Statistical Analysis

Categorical variables were described in terms of integer numbers and percentages. Numerical traits were depicted with their mean, median, standard deviation, and minimum–maximum values. The Person Chi-squared test of independence was performed for descriptive between-group purposes. A binary logistic regression model was carried out in order to estimate adjusted odds ratios for clinical conditions, controlling for age and gender.

A level of *p* < 0.05 was deemed statistically significant. All statistical procedures were performed using Statistica™, release 14 (TIBCO Software Inc., Palo Alto, CA, USA).

## 5. Conclusions

Hypersensitivity to NSAIDs is a significant and difficult diagnostic problem in pediatric allergy. At present, it is an issue as pressing as allergies to beta-lactam antibiotics. The most common causative drug among NSAIDs in children is ibuprofen. The most common manifestations of hypersensitivity to ibuprofen in children are acute urticaria and angioedema. Currently, diagnostic methods are largely limited to the oral challenge test.

Going forward, in future research, it will be important to take into account cofactors such as vitamin D levels and viral infections.

## Figures and Tables

**Table 1 pharmaceuticals-16-01237-t001:** Baseline characteristics of the study cohort (*n* = 56 *).

Analyzed Trait	Statistical Parameter **	
	*n* M (SD)	% Me (Q_1_–Q_3_)
History of allergy	26	46.4
Family allergy	14	25.0
Gender:		
Female	25	44.6
Male	31	55.4
Age (y)	10.7 (4.4)	11 (6–15)
Comorbidities (chronic):		
Arterial hypertension	1	1.8
Asthma	8	14.3
Psoriasis	2	3.6
Allergic rhinitis	11	19.6
Severity:		
Anaphylactic shock	2	3.6
Angioedema/urticaria	47	83.9
maculopapular rash	7	12.5
Time of reaction:		
Immediate	31	55.4
Delayed	19	33.9
Late	6	10.7
Other medicines:		
Herbal syrups	12	21.4
Local antiseptics	5	8.9
Other	1	1.8
SPT (skin prick test)	0	0.0
Provocation test positive	17	30.4
EBV IgG	22	39.3
Vitamin D deficit (<20 ng/mL)	31	55.4

(* Missing data were deleted case-wise, if applicable. ** For categorical variables: *n*—integer number; %—percentage. For numerical traits: M—mean; SD—standard deviation; Me—median; min.-max.—minimum-to-maximum values).

**Table 2 pharmaceuticals-16-01237-t002:** Provocation test result in the studied patients by history of allergy.

History of Allergy	Provocation Test Result				Total	*p*-Value
	Positive		Negative			
	*n*	%	*n*	%		
Yes	17	65.4	9	34.6	26	<0.0001
No	0	0.0	30	100.0	30	
Overall	17	30.4	39	69.6	56	

**Table 3 pharmaceuticals-16-01237-t003:** Provocation test result in the studied patients by family allergy.

History of Allergy	Provocation Test Result				Total	*p*-Value
	Positive		Negative			
	*n*	%	*n*	%		
Yes	10	71.4	4	28.6	14	<0.0001
No	5	12.8	34	87.2	39	
Overall	15	28.3	38	71.7	53	

OR = 17.00 (95%CI: 3.82–75.57).

**Table 4 pharmaceuticals-16-01237-t004:** Provocation test result in the studied patients by gender.

Gender	Provocation Test Result				Total	*p*-Value
	Positive		Negative			
	*n*	%	*n*	%		
Girls	11	44.0	14	56.0	25	=0.0461
Boys	6	19.3	25	80.7	31	
Overall	17	30.4	39	69.6	56	

**Table 5 pharmaceuticals-16-01237-t005:** Provocation test result in the studied patients by vitamin D status.

Vitamin D Status	Provocation Test Result				Total	*p*-Value
	Positive		Negative			
	*n*	%	*n*	%		
Normal	3	12.5	21	87.5	24	=0.0092
Lowered	14	45.2	17	54.8	31	
Overall	17	30.9	38	69.1	55	

## Data Availability

Data is contained within the article.

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
