# Peer review of "Practical Approach to Hypersensitivity to Nonsteroidal Anti-Inflammatory Drugs (NSAIDs) in Children"

_pharmaceuticals, 2023, doi:10.3390/ph16091237_

Round 1

Reviewer 1 Report

Please add when NSAIDs are used by the children or in the study group age (The current analysis is restricted to 56 children, aged 4 to 18 years). What is the restriction for the use of NSAIDs, for example below the age of xx is not recommended or contraindicated. In addition, please clarify these for various NSAIDs as there are several compounds in this class of drugs. The authors have mentioned two mostly used and those can be given with more information about the age restriction per FDA approved and indicated on the drug labels.

Is hypersensitivity restricted to oral? or can be any other administration route, such as topical or rectal. Please elaborate on factors that can play a role in hypersensitivity response or not.  In this line, please also consider sex-related differences in hypersensitivity. Did the authors identify any difference in this regard? Did they sub-analyze data or not? In either case, please elaborate on the sex-related response to drug hypersensitivity in general and also concerning NSAIDs. If data are not available for children, please add information if the literature has some data for adults.

The authors have mentioned a couple of limitations of this study including the small sample size. Do the authors consider the high internal validity and low external validity of this study? In other words, can the data be generalized?

Please add how these findings can be used in the clinic. Knowing the hypersensitivity related to NDAIDs in children, what can be done? please elaborate on this considering patients, parents, and clinicians.

The authors have mentioned that no other better test is available, but can they elaborate as to what can be invented or created for the future for predicting hypersensitivity to NSAIDs n children and adolescents? Genetic test? or other evaluations?

To avoid any potential hypersensitivity do the authors recommend switching to other tools for reduction of e.g., pain?

Is there any cross-sensitivity between NSAIDs and other drugs of known reports in the literature+ if so, please add. For example, sensitivity to penicillin can predict a higher potential for having a higher sensitivity to NSAIDs.

Please add treatment strategies for severe hypersensitive reactions to NSAIDs when they occur in children. can this be life-threatening?

The authors have added that they have obtained consent from participants. Does this mean for all or those who could give consent or their parents or guardians? Considering the age group in this study (4 to 18), ethical approvals and double parenting consents look essential. Please add the information.

Please proofread the text. There are some editorial and grammatical errors, punctuation, etc.

Author Response

Dear reviewer, thank you for your comments. Below you will find our answers.

Q1 Please add when NSAIDs are used by the children or in the study group age (The current analysis is restricted to 56 children, aged 4 to 18 years). What is the restriction for the use of NSAIDs, for example below the age of xx is not recommended or contraindicated. In addition, please clarify these for various NSAIDs as there are several compounds in this class of drugs. The authors have mentioned two mostly used and those can be given with more information about the age restriction per FDA approved and indicated on the drug labels.

Resp: Nonsteroidal anti-inflammatory drugs are the most frequently given drugs in infection as pain-killers, anti-inflammatory or antipyretic drugs.  In accordance with the FDA recommendations and the characteristics of medicinal products, paracetamol, ibuprofen or metamizole are allowed for use in the youngest children as analgesic, anti-inflammatory or antipyretics.  Acetylsalicylic acid is approved for use in children over 12 years of age,  due to probable appearance of Reye’s syndrome. Due to the above, in the presented study, cases of hypersensitivity after the above-mentioned four drugs were analyzed. Of course, in special situations, such as severe pain, especially of rheumatoid origin, or no alternative, naproxenum can be used over 5 years of age. We usually use naproxen as alternative drug in patients with confirmed allergy to paracetamol and ibuprofen.  In our study, as mentioned in inclusion criteria, patients with diagnosed chronic disease (including rheumatology) were excluded from the assessment.

This information was added to the main text in the introduction section in red.

Q2. Is hypersensitivity restricted to oral? or can be any other administration route, such as topical or rectal. Please elaborate on factors that can play a role in hypersensitivity response or not.  In this line, please also consider sex-related differences in hypersensitivity. Did the authors identify any difference in this regard? Did they sub-analyze data or not? In either case, please elaborate on the sex-related response to drug hypersensitivity in general and also concerning NSAIDs. If data are not available for children, please add information if the literature has some data for adults.

Resp: Hypersensitivity to NSAIDs is not limited  only to oral administration of the preparations. This is the most common route of administration, but there are cases of hypersensitivity with rectal and/or intravenous administration. In our group of patients, in 48 patients symptoms suspected of hypersensitivity occurred after oral administration, in 4 cases after rectal administration (suppository) and in 2 cases after intravenous administration during infection.

In a recent study of Kuhl et al analyzing cofactors of drug hypersensitiivity  a positive association of antibiotic hypersensitivity with obesity was found and also age was negatively associated with arterial hypertension, female sex, elevated immunoglobulin E, and allergic rhinitis. Hypersensitivity to NSAIDs was associated with atopic dermatitis, elevated IgE, and arterial hypertension.  Our study showed previous history of allergy in studied patients and also previous history of allergic family conditions, were risk factors for drug allergy in studied patients, p-value 0,001.  What is more, in the pediatric population, especially in children <7 years of age, most drugs are administered in the form of syrups or suspensions. In the drug prepared in this way, apart from the active substance, other ingredients are also present, such as, for example, a flavoring agent, an emulsifier, a stabilizer, a preservative, and so on. In practice, any of these substances could cause a hypersensitivity reaction. Therefore, during the diagnostics, during the oral challenge test, we used a powdered and measured part of the drug, with the aim of administering the purest form of the drug. This information was added to the main text in the discussion  section in red.

 In the presented study hypersensitivity to NSAIDs was confirmed statistically more often in girls compared to boys (P=0.046). Data on the relationship between drug allergy in children and gender are currently lacking in the literature. Eaddy et al in his study showed drug allergy to be more frequently reported in adult females compared with adult males. Adult females are also more likely to have drug-induced anaphylaxis. Studies in children suggest the reverse picture, with prepubertal males more likely to have drug allergy and drug-induced anaphylaxis than prepubertal girls. Possible explanations for women's predisposition to drug allergies are rather multifactorial and may include greater exposure to antibiotics or drugs, X-linked genetic factors, epigenetic changes, and divergent hormonal interactions with immune cells.  Also in study of Sousa-Pint et al the prevalence of self-reported drug allergy was higher in female patients, adults, and inpatients. More studies are needed in this field. This information was added to the main text in red.

Eaddy Norton A, Broyles AD. Drug allergy in children and adults: Is it the double X chromosome? Ann Allergy Asthma Immunol. 2019 Feb;122(2):148-155. doi: 10.1016/j.anai.2018.11.014. Epub 2018 Nov 20. PMID: 30465863.

Kühl J, Bergh B, Laudes M, Szymczak S, Heine G. Cofactors of drug hypersensitivity-A monocenter retrospective analysis. Front Allergy. 2023 Jan 6;3:1097977. doi: 10.3389/falgy.2022.1097977. PMID: 36686964; PMCID: PMC9854260.

Q3 The authors have mentioned a couple of limitations of this study including the small sample size. Do the authors consider the high internal validity and low external validity of this study? In other words, can the data be generalized?

Resp: The high internal validity of the present study is in fact very satisfying and incontestable. However, taking into account the discussed limitations of the study, a reliable estimation of its internal validity is at the Investigators’ discretion. The Investigators are aware of those limitations while being clearly inclined to believe that the research topic at issue is undoubtedly worth continuing in a larger group of subjects.

Q4. Please add how these findings can be used in the clinic. Knowing the hypersensitivity related to NDAIDs in children, what can be done? please elaborate on this considering patients, parents, and clinicians.

Resp: Our results indicate that low serum vitamin D status may be a risk factor for hypersensitivity to NSAIDs. According to pediatric recommendations, each child should be supplemented with vitamin D. However, it is worth taking into account the type of diet in the child and the individual absorption of vitamin D in each patient and testing the level of vitamin D, for example, at annual check-up visits. In addition, in children with a history of hypersensitivity, even if it was not confirmed by an oral challenge test, it is worth using drugs in the form of tablets to avoid additional potentially allergenic substances. In patients with confirmed hypersensitivity to one drug from the group of NSAIDs, it is advisable to perform an oral challenge test with an alternative drug in order to find a safe drug for the child.

This information was added to the main text in red.

Q5. The authors have mentioned that no other better test is available, but can they elaborate as to what can be invented or created for the future for predicting hypersensitivity to NSAIDs in children and adolescents? Genetic test? or other evaluations?

Resp: To date, the gold standard of management in the case of suspected hypersensitivity to NSAISs remains an oral challenge test with a potentially causative or alternative drug. The available tests, such as the basophil activation test and the blast transformation test, have little use for NSAISs and are not standardized. Perhaps in the future it will be possible to create a genetic profile of a patient with a high risk of hypersensitivity to drugs, but this area of ​​knowledge is still in the field of research.

Q6 To avoid any potential hypersensitivity do the authors recommend switching to other tools for reduction of e.g., pain?

Resp: NSAIDs are used as anti-inflammatory, antipyretic and analgesic drugs. When treating children, the rule is to use the least necessary amount of control medication. In the case of fever, the use of cooling compresses is often necessary, but it does not always bring the desired effect, hence the need for pharmacotherapy. Similarly, other methods of analgesic treatment, such as aromatherapy, massage or acupuncture, are also used in pain management in many countries and probabely should could be routinely used. 

Q7. Is there any cross-sensitivity between NSAIDs and other drugs of known reports in the literature+ if so, please add. For example, sensitivity to penicillin can predict a higher potential for having a higher sensitivity to NSAIDs.

Resp:  According to current knowledge, it is known that hypersensitivity to one agent may predispose to the development of hypersensitivity to other agents. Much is known about cross-allergy between drugs of one class, such as in the group of beta-lactams or in the group of NSAIDs. To our knowledge, there are no studies available in the literature on cross-sensitivity between  other drugs and NSAIDs in children.

Q8. Please add treatment strategies for severe hypersensitive reactions to NSAIDs when they occur in children. can this be life-threatening?

Resp: Hypersensitivity reactions can be severe in both children and adults and may be life threatening. In the case of NSAIDs, severe hypersensitivity reactions are rare, both in the form of anaphylactic shock or  severe cutaneous drug allergy reactions (SCARS) as Stevens-Johnson syndrome (SJS), toxic epidermal necrolysis (TEN) , drug reaction with eosinophilia and systemic symptoms (DRESS).  In pediatric population, the most common SCAR phenotypes were reported to be SJS/TEN, DRESS . The main reported culprit drugs for SJS/TEN in children are antibiotics such as beta-lactams and sulfonamides.  Some reports suggest,  that SCARs are less prevalent and have a better prognosis being less associated with comorbidities in the pediatric population compared with an adult population. According to International Consensus on drug allergy, drug provocation tests in these cases are contraindicated. In such cases, drugs are used only with strict indications and under the supervision of a doctor. Drugs with presumed low cross-reactivity are selected and administered in a hospital setting. There are still no established standards of care for this group of patients.   The most important step in management is an immediate cessation of all potentially implicated drugs and aggressive supportive care. In the case of anaphylactic shock, the basic treatment is intramuscular adrenaline, followed by an antihistamine and glucocorticosteroids

Systemic corticosteroids have been accepted as the standard treatment for improving clinical symptoms of DIHS/DRESS at the acute phase; however, the evidence for this is largely through case reports and case series. Cyclosporine is the most effective therapy for the treatment of SJS, and a combination of intravenous immunoglobulin (IVIg) and corticosteroids is most effective for SJS/TEN overlap and TEN.

Q9. The authors have added that they have obtained consent from participants. Does this mean for all or those who could give consent or their parents or guardians? Considering the age group in this study (4 to 18), ethical approvals and double parenting consents look essential. Please add the information.

Resp: In accordance with the legal requirements in our country, in the presented study, informed consent to the diagnostic procedures and participation in the study was obtained both from parents/guardians of all children and additionally fromall children over 16 years of age. This information was skipped by mistake and now has been added to IRB statement

Q10  Please proofread the text. There are some editorial and grammatical errors, punctuation, etc.

Resp: The  English editing was now performer by a native speaking person.

Reviewer 2 Report

Dear Editor

many thanks for asking me to review this manuscript. Authors aimed to assess  the real-life prevalence, patient profile, and clinical presentation of drug hypersensitivity to NSAIDs in children referred to their clinic after an incidence of adverse event during treatment, verified by drug challenge test.

Please, specify inclusion and exclusion criteria of the patients.

Has been the power sample size calculated?

Did the study requested IRB approval?

I also suggest to improve the quality of the literature search, also addin a recent paper focusing on drug-allergy in childre. Please, cite: Cravidi C, et al. Drug Allergy in children: focus on beta-lactams and NSAIDs. Acta Biomed. 2020;91(11-S):e2020008. Published 2020 Sep 15. doi:10.23750/abm.v91i11-S.10312

Dear Editor

many thanks for asking me to review this manuscript. Authors aimed to assess  the real-life prevalence, patient profile, and clinical presentation of drug hypersensitivity to NSAIDs in children referred to their clinic after an incidence of adverse event during treatment, verified by drug challenge test.

Please, specify inclusion and exclusion criteria of the patients.

Has been the power sample size calculated?

Did the study requested IRB approval?

I also suggest to improve the quality of the literature search, also addin a recent paper focusing on drug-allergy in childre. Please, cite: Cravidi C, et al. Drug Allergy in children: focus on beta-lactams and NSAIDs. Acta Biomed. 2020;91(11-S):e2020008. Published 2020 Sep 15. doi:10.23750/abm.v91i11-S.10312

Author Response

Q1. Please, specify inclusion and exclusion criteria of the patients.

Resp: Inclusion criteria:  We included 56 children, aged 4 to 18 years with history of incidence of adverse reac-tions during treatment with NSAIDs in the last 6 months from our Pediatrics and Allergy Out-patient Clinic. 

Exclusion criteria: Patients with chronic diseases requiring a prolonged drug intake were excluded from the study. Patients with severe drug allergy reactions were excluded from the study as severe drug allergy reaction is beeing a contraindication for drug provocational test.

Any mental disorders limiting patient contact and compliance were also an exclusion criteria.

Inclusion and exclusion criteria were extended and added to the text in red.

Q2 Has been the power sample size calculated?

Resp: Yes, the statistical power was appraised. The Authors would like to express, considering all the statistical analyses performed, the said power did exceed the threshold of 0.9 for each statistical procedure being fitted.

Q3. Did the study requested IRB approval?

Resp. The study reqested IRB approval.  The study was approved by the Medical Ethics Committee of the Medical University of Lodz; RNN/147/18/KE. All parents or legal guardians gave their oral and written consent for the evaluation of data from medical documentation of their children.

This information was skiped by mistake and now has been uploaded in the IRB statement.

Q4. I also suggest to improve the quality of the literature search, also adding a recent paper focusing on drug-allergy in childre. Please, cite: Cravidi C, et al. Drug Allergy in children: focus on beta-lactams and NSAIDs. Acta Biomed. 2020;91(11-S):e2020008. Published 2020 Sep 15. doi:10.23750/abm.v91i11-S.10312

Resp. Thank You for your suggestion, The above paper has been cited.

Round 2

Reviewer 1 Report

The authors have reflected upon the comments-suggestions and revised the manuscript accordingly. Thanks. There are no further comments.